# Combining In Silico and In Vitro Studies to Evaluate the Acetylcholinesterase Inhibitory Profile of Different Accessions and the Biomarker Triterpenes of *Centella asiatica*

**DOI:** 10.3390/molecules25153353

**Published:** 2020-07-24

**Authors:** Nor Atiqah Jusril, Ain Nur Najihah Muhamad Juhari, Syahrul Imran Abu Bakar, Wan Mazlina Md Saad, Mohd Ilham Adenan

**Affiliations:** 1Faculty of Applied Sciences, Universiti Teknologi MARA, Shah Alam 40450, Selangor Darul Ehsan, Malaysia; atiqahjusril93@gmail.com (N.A.J.); ainnrnjhh@gmail.com (A.N.N.M.J.); syahrulimran@uitm.edu.my (S.I.A.B.); 2Atta-ur-Rahman Institute for Natural Product Discovery (AuRIns), Level 9, FF3 Puncak Alam Campus, Universiti Teknologi MARA, Puncak Alam 42300, Selangor Darul Ehsan, Malaysia; 3Centre of Medical Laboratory Technology, Faculty of Health Sciences, Universiti Teknologi MARA, Puncak Alam 42300, Selangor Darul Ehsan, Malaysia; wanmaz755@uitm.edu.my; 4Universiti Teknologi MARA Pahang Branch, Bandar Tun Abdul Razak 26400, Jengka, Pahang Darul Makmur, Malaysia

**Keywords:** *Centella asiatica*, acetylcholinesterase, triterpenes, molecular docking, in silico, in vitro

## Abstract

Alzheimer’s disease (AD) is a neurodegenerative disease and the most cause of dementia in elderly adults. Acetylcholinesterase (AChE) is an important beneficial target for AD to control cholinergic signaling deficit. *Centella asiatica* (CA) has proven to be rich with active ingredients for memory enhancement. In the present study, the chemical profiling of three accession extracts of CA namely SECA-K017, SECA-K018, and, SECA-K019 were performed using high-performance liquid chromatography (HPLC). Four biomarker triterpene compounds were detected in all CA accessions. Quantitative analysis reveals that madecassoside was the highest triterpene in all the CA accessions. The biomarker compounds and the ethanolic extracts of three accessions were investigated for their acetylcholinesterase (AChE) inhibitory activity using Ellman’s spectrophotometer method. The inhibitory activity of the triterpenes and accession extracts was compared with the standard AChE inhibitor eserine. The results from the in vitro study showed that the triterpene compounds exhibited an AChE inhibitory activity with the half-maximal inhibitory concentration (IC_50_) values between 15.05 ± 0.05 and 59.13 ± 0.18 µg/mL. Asiatic acid was found to possess strong AChE inhibitory activity followed by madecassic acid. Among the CA accession extracts, SECA-K017 and SECA-K018 demonstrated a moderate AChE inhibitory activity with an IC_50_ value of 481.5 ± 0.13 and 763.5 ± 0.16 µg/mL, respectively from the in silico docking studies, it is observed that asiatic acid and madecassic acid showed very good interactions with the active sites and fulfilled docking parameters against AChE. The present study suggested that asiatic acid and madecassic acid in the CA accessions could be responsible for the AChE inhibitory action and could be used as markers to guide further studies on CA as potential natural products for the treatment of AD.

## 1. Introduction

Alzheimer’s Disease (AD) is a neurodegenerative disease which is the common cause of dementia and is mainly differentiated by progressive deterioration of memory and cognition [1]. It is characterized by low levels of the neurotransmitter acetylcholine (ACh), neuro-inflammation, and oxidative stress in the brain region [2]. There are millions of people worldwide with AD and dementia and the World Health Organization (WHO) estimates the number of people with AD is rising rapidly [3]. Nowadays, about 24 million of peoples are affected by AD and it is predicted that this number will quadruple by 2050 [4]. Previous studies performed on AD patients found an altered cholinergic activity, which resulted in disruption cognitive and functional symptoms [5]. AD is reported to affect older people of 65 years old and above, resulting in memory and behavior impairment [6]. One of the most promising targets for AD treatment is by suppressing the acetylcholinesterase (AChE) activity in the brain to ameliorate the cognitive ability [7].

AChE is a key of enzyme that plays important roles in cholinergic transmission by hydrolyzing the neurotransmitter acetylcholine (ACh) [8]. Inhibition of AChE in the cholinergic framework is intimately linked with the therapy of neurodegenerative-related diseases [9]. Therefore, by inhibiting AChE, the levels of this neurotransmitter can be elevated and thus improve the learning and memory functions. AChE inhibitors can be divided into two categories: reversible and irreversible. In general, reversible inhibitors have therapeutic applications, while irreversible AChE inhibitors are associated with toxicity effects [10]. The major drugs currently available for the treatment of neurodegenerative disorders are donepezil, tacrine, eserine, rivastigmine, huperzine A, and galantamine, which have a number of side effects [11,12]. Meanwhile, natural products from medicinal plants are known for their inherent benign safety and effects [13]. Therefore, it would be beneficial to discover other inhibitors of AChE with more selectivity, enhance the bioavailability problems, and produce fewer adverse effects to the AD patients. The finding of inhibitors of AChE from natural products is increasing and proving to be a promising source of useful AChE inhibitors [14,15].

There are a number of active compounds with good cholinesterase activity that have been isolated from medicinal plants [16]. *Centella asiatica* (L.) Urban (CA) is a herbal plant from the Apiaceae family which is native to South and Southeast Asian countries including Sri Lanka, India, China, Malaysia, and Indonesia [17]. In Malaysia, it is also known as pegaga and is one of the medicinal plants traditionally used for brain and nerve cell revitalization [18]. The major bioactive chemical compounds in CA are alkaloids, triterpenes, flavonoids, volatile oils, and glycosides [19]. Among the different classes of natural products, the triterpenoids are the most diverse class of organic compounds. The primary bioactive compounds in CA are madecassoside, asiaticoside, madecassic acid, and asiatic acid, which belong to the class of triterpenes [20,21] (Figure 1). Previous study reported that these compounds have been utilized as biomarker components for the quality assessment of raw materials and herbal products of CA. However, environmental factors such as climate (temperature, humidity, wind, and light), location, growth, and soil fertility could affect the chemical components in natural plants [20,22]. Devkota et al., (2015) reported that the environmental conditions such as light exposure, fertilizer, and soil type could affect the triterpenes content in CA [23]. Therefore, a proficient qualitative and quantitative analysis is needed to ensure their efficacy, quality, and safety. The neuropharmacological value of CA on neuroprotection and the regeneration of the peripheral nervous system has been widely investigated [21]. Therefore, it is more likely that the presence of triterpenes in CA may be effective against AD [11].

Molecular docking is a tool in computer aided drug designing (CADD) which is used to study the binding interaction between the potential inhibitors, known as ligands, and targeted enzymes [24]. The X-ray crystal structures of AChE from various species can be found in the protein data bank (PDB) [25]. The identification of residues that are responsible for inhibitory activity will lead to potential for the synthesis of agents with a high efficacy of biological action [26].

In Malaysia, there are more than 15 accessions of CA with each having variation in their bioactive constituents [27]. Zainol et al., (2003) observed two accessions of CA that have variation in their active constituents in different parts of plants such as the root, petiole, and leaf of CA and showed high antioxidant activities [28]. In a different study, 14 accessions of CA were identified and from HPLC fingerprinting analysis. It was observed that madecassic acid was highest in one of the accessions [29]. In other findings, five accessions of CA from India showed variations in secondary metabolites due to the difference in altitude and geographical location of the plant [30]. In China, there were 14 accessions of CA collected from different locations with a different latitude, longitude, and collection time, and they observed the variation in chemical composition and genetic diversity [31]. Although many pharmacological effects of CA have been reported, the AChE inhibitory activity among accessions of CA in Malaysia has not been fully investigated. Therefore, the aim of this study is to perform high performance liquid chromatography (HPLC) fingerprinting analysis and AChE inhibitory activity of triterpenes among the CA accessions. In addition, a molecular docking study was also carried out to investigate the binding interaction of targeted compounds against AChE.

## 2. Results

### 2.1. Characterization of CA Accessions

The morphological character analysis carried out for the three accessions of CA, namely CA-K017, CA-K018, and CA-K019, in relation to their leaf shape, color, and diameter size is shown in Table 1. It was observed that leaves of both accessions, CA-K017 and CA-K019, were light green while CA-K018 was dark green. The leaf margin of CA was divided into three different groups. These were crenate (surface with rounded edge), crenulate (having a wavy edge), and crenate with dentate base (tooth-shape projection) [32]. Accession CA-K017 was found to have a crenate with dentate based leaf margin; meanwhile a crenate with a rounded edge leaf margin was detected in accession CA-K018. However, CA-K019 accession had a crenulate margin. The length of leaf when expressed in centimeters (cm), showed the CA-K018 accession had the longest leaf length, followed by CA-K019 and CA-K017 as shown in Table 1.

### 2.2. Percentage Yield of Extraction

In the present study, three accessions of CA were extracted with 95% denatured ethanol, and percentage of yield was calculated. The extraction yield of the three extracts of CA accessions designated as SECA-K017, SECA-K018, and SECA-K019 is shown in Table 2. Results showed that SECA-K018 gave the highest extraction yield compared to SECA-K017 and SECA-K019.

### 2.3. HPLC Analysis

HPLC identification and quantification of four triterpene compounds of SECA accessions were made based on their retention time (RT) and ultraviolet-visible (UV-VIS) spectra at 206 nm (Table 3). Figure 2A–D show a chemical profile of CA accessions and the reference standards of madecassoside, asiaticoside, madecassic acid, and asiatic acid. Simultaneous quantification of triterpenes in CA has also been achieved using this method. It has been reported that asiaticoside and madecassoside are the most abundant triterpenes found in CA; however, the concentrations of these compounds may vary depending on geographical factors such as environment, origin, harvesting time, as well as the extraction method used [20]. In the present study, the content of madecassoside was found to be the highest among the accessions, followed by asiaticoside. This finding is in agreement with previous the study on the CA extract from Thailand containing about 80% triterpenoids glycosides such as madecassoside (53.1%), and asiaticoside (32.3%) [33]. The recent report from L. Yulianti et al., (2017) has confirmed that concentrations of madecassoside and asiaticoside in CA were higher than madecassic acid and asiatic acid [34].

In the present study, the detection of four triterpenes in CA was in line with Hashim et al., (2011) where four triterpenes were also present in their CA accessions [35]. Aziz et al., (2007) observed that the leaves contain the highest composition of triterpenes compared to roots. In addition, the phenotype of CA with a smooth leaf had the highest level composition of madecassoside and asiaticoside compared to those with a fringed leaf [36]. These findings concluded that different locations of plants could affect the triterpenes composition. Therefore, there is need for continuous fingerprinting analysis to standardize the herbal plants to promote the development of phyto-drugs or functional foods. This may also improve the yield of active constituents of the plants that will give optimal triterpenes, which can be used as biomarkers.

Plant-derived secondary metabolites such as triterpenoids, alkaloids, and flavonoids have proved their medicinal properties for the treatment of neurodegenerative disorders, cancer, cardiovascular diseases, and skin diseases [11]. Therefore, it is more likely that triterpenoid presence in CA may be one of the promising inhibitors against AD. It has been reported that a functional moiety in lupine-type triterpenoids has shown a potential effect against neurodegenerative disorders [37]. In addition, literature supplies numerous reports on the enzyme that can be inhibited by pentacyclic triterpenoids, which reveals the ability of these compounds to easily bind on multiple targets based on hydrophobic interaction with an enzyme’s domain [38].

### 2.4. Acetylcholinesterase Inhibitory Activity

Four biomarker compounds of CA and three extracts of accessions of CA (SECA-K017, SECA-K018, and SECA-K019), were tested for in vitro AChE inhibitory activity. All samples inhibited AChE activity in a dose-dependent manner. The results expressed as half maximal inhibitory concentration (IC_50_) values, are shown in Table 4. In our study, the AChE inhibitory activity exhibited by CA accessions could be associated with the content of triterpenes as shown by the HPLC analysis. Asiatic acid and madecassic acid showed a notable AChE inhibitory effect with IC_50_ values of 15.05 ± 0.06 µg/mL and 17.83 ± 0.05 µg/mL, respectively, lower than the CA accessions and others triterpenes, which corroborates their competitive, selective, and reversible affinity for AChE [39]. Meanwhile, SECA-K017 and SECA-K018 extracts demonstrated a moderate AChE inhibitory activity with an IC_50_ value of 481.5 ± 0.13 and 763.5 ± 0.16 µg/mL, respectively. The other accession, SECA-K019, did not show AChE inhibitory activity. Therefore, SECA-K017 and SECA-K018 extracts demonstrated inhibitory action against the AChE enzyme. To the best of our knowledge, there are no studies for the evaluation of different CA accessions on AChE inhibitory activity that have been done.

It has been reported that a glycosidic compound exhibited a potent neuroprotective activity, improved microglial activation and behavioral dysfunction [40]. Previous studies proved that asiatic acid could reduce the level of corticosterone in a rat’s brain, which ameliorates the content of monoamine neurotransmitters and increases the function of hypothalamic-pituitary adrenal for its antidepressant effects [41]. Neagu et al., (2018) reported that asiatic acid and madecassic acid inhibited the AChE enzyme more effectively than other triterpenes as it could support the evidence for the use of CA for cognitive function improvement. Herbal medicines have been used for the treatment of memory and cognitive functions [42]. According to S. Bhadra et al., (2016) asiatic acid, a triterpenoid component of CA, was found to have an inhibitory effect against the AChE enzyme [9]. N. Omar et al., (2019) reported that asiatic acid that was present in CA could cross the blood–brain barrier (BBB) and maintained in the tight junction of the BBB [24]. Furthermore, it has been reported that the combination of asiatic acid and madecassic acid could induce neuronal differentiation and neurofilament [39]. However, the comparison between triterpenes compounds and different accessions of CA are still undetermined. In addition, methanolic extracts of CA species from India were demonstrated to exert a potent cholinesterase inhibition activity, in vitro free radical scavenging, and the improvement of scopolamine-induced amnesia activity [43]. The present study demonstrated that SECA-K017 and SECA-K018 containing four triterpene compounds showed AChE inhibitory activity. However, SECA-K019 did not show AChE inhibitory activity probably due to the lack of madecassic acid content. In an attempt to confirm the bioactivity of a CA extract, molecular docking was conducted to recognize the binding interactions of triterpene compounds in the catalytic site of the crystal structure of AChE (PDB: 4EY7).

### 2.5. Molecular Docking

Molecular docking is a tool to predict the binding interaction of ligands towards targeted proteins as well as giving the binding affinity of small compounds [44]. In order to confirm the in vitro results and to find out potential residues towards the active site of targeted enzyme, the ligand–enzyme binding interactions between CA active compounds and the AChE enzyme were evaluated using AutoDock 4.2 [45]. The results of the estimated binding interaction energies with the active site of AChE are presented in Table 5.

The in vitro studies showed that asiatic acid and madecassic acid have the highest inhibitory activity with the lowest binding energy of −10.27 and −8.7 Kcal/mol, respectively, compared with other triterpenes, but lower than eserine against AChE. The docking results obtained for asiatic acid and madecassic acid suggest that the present hydroxyl group at the C-1, C-2, and C-3 position theoretically improves the AChE inhibition of the 4EY7 enzyme. The ligand-enzyme binding interaction representations of the best conformation of the complexed active sites’ interaction of AChE with eserine (a), asiatic acid (b), and madecassic acid (c) are presented in Figure 3. The molecular interactions between triterpene compounds and active sites of the AChE protein were analyzed in terms of hydrogen bonding and π-π stacking interactions. Tyr337 and Trp86 are the important residues of AChE as these amino acids function to maintain the geometry of the binding gorge and provide electrostatic balance [46]. Previous studies have also highlighted the importance of Tyr337 and Trp86 amino acids in the binding activity of protein 4EY7 [47]. Regarding the hydrogen bonding, eserine showed the highest stability due to conventional hydrogen bond interaction with Phe295 (2.51Å), while asiatic acid presented three strong hydrogen bonding interactions with His447 (1.24 Å), Tyr337 (3.72 Å), and Arg296 (1.97 Å) residues. In addition, madecassic acid demonstrated strong hydrogen bonding interaction with Tyr341 (4.31 Å), Phe295 (2.64Å), and Arg296 (2.29 Å). The bond lengths of respected interactions between compounds and amino acids were determined (Appendix A, Figure A1). Furthermore, eserine and asiatic acid compounds have a similar bonding interaction according to carbon–hydrogen bonding with Trp286 and Gly121. Meanwhile, madecassic acid showed carbon–hydrogen bonding with Ser293 (1.24 Å).

Additionally, the ligand-enzyme binding interactions are stabilized by the presence of hydrophobic alkyl, hydrophobic π-alkyl, and hydrophobic π-sigma interactions with the compounds inside AChE as shown in Figure 3b,c,e [48]. The theoretical results obtained by molecular docking for asiatic acid and madecassic acid are in agreement with in vitro assay. According to the molecular interactions shown in the docking analysis, the asiatic acid and madecassic acid that is present in CA may be represented as a potential inhibitor based on its very good interaction due to strong hydrogen bonds and hydrophobic interactions.

## 3. Chemicals and Reagents

### 3.1. Chemicals and Reagents

All solvents were HPLC grade, purchased from Merck (Darmstadt, Germany). The pure chemical standards of asiaticoside, madecassoside, asiatic acid, and madecassic acid were purchased from Chemfaces (Wuhan, Hubei, China). Sodium monobasic sulfate and sodium dibasic sulfate were purchased from Merck (Darmstadt, Germany), and acetylcholinesterase from electrophorus electricus, 5,5′-Dithiobis (2-nitrobenzoic acid), and acetylcholine iodide were obtained from Sigma–Aldrich (St. Louis, MO, USA).

### 3.2. Plant Characterization and Extraction

Three accessions of CA of different characteristics were collected from different geographical localities in Malaysia. These three accessions were denoted as CA-K017, CA-K018, and CA-K019. Matured plants were examined and measured based on morphological characteristics. The morphological characteristics measured were color and leaf margin. All the parameters were recorded in triplicates. The whole plant was washed, cleaned, and oven-dried at 40 °C. The powdered plant materials were extracted using 95% denatured alcohol at a room temperature for 72 h. The yields of the extracts were calculated. These extracts were designated as standardized extracts CA (SECA) of SECA-K017, SECA-K018, and SECA-K019. Voucher specimens were prepared and deposited in the Faculty of Applied Sciences, UiTM Shah Alam (Selangor, Malaysia) for future reference.

### 3.3. HPLC Fingerprinting Analysis

An HPLC system (Agilent, Santa Clara, CA, USA) equipped with a diode-array UV-vis detector and C18 HPLC column (15 cm × 4.6 mm i.d., 5 µm) (Supelco, Bellefonte, PA, USA) was used. The mobile phase consists of water (A) and acetonitrile (B) using a gradient elution program for 55 min with a flow rate of 1 mL/min and a detection wavelength at 206 nm for analysis as described by previous literature with slight modifications (Table 6) [49]. About 10 mg samples were suspended in methanol:water (7:3) and filtered through a 0.2 µm polyvinylidene fluoride (PVDF) membrane syringe filter, prior to the HPLC analysis. Standard solutions were prepared in methanol at concentration 1000 µg/mL. An appropriate volume of each standard solution was mixed and diluted with methanol to obtain 5 concentrations ranging from 10–1000 ppm to obtain calibration curves for quantitative analysis. The relative amount of the compound was expressed as milligram per gram extract.

### 3.4. Acetylcholinesterase Inhibitory Activities

The AChE inhibitory activity was evaluated using Ellman’s method, as reported previously [48]. Each sample (20 µL of 5 mg/mL in DMSO) was dispensed in triplicate into a 96-well microplate and mixed with 190 µL of DTNB, 20 µL of substrate (ATCI). The control wells contained 2% of DMSO instead of the extract. The enzymatic activity was measured at 412 nm every 30 s intervals for 3 min. The buffer solution was used as negative control. 

The percentage inhibition (%I) of each sample and the positive control (physostigmine) was calculated using the formula:%I: [(A_c_ − A_s_)/(A_c_)] × 100(1)
where (%I): Percentage inhibitionA_c_: Absorbance of negative controlA_s_: Absorbance of sample

### 3.5. Molecular Docking

The molecular docking study of triterpene compounds was performed to evaluate the binding interaction mode in the active site of the AChE enzyme (4EY7) [50]. The 3D structures of the triterpenes were drawn with the Chemdraw program and optimized to confirm the potential energy surfaces. The dockings for the set of ligands with the respective enzyme were performed using Autodock 4.2. The 3D model of the AChE enzyme was downloaded from Research Collaboratory for Structural Bioinformatics (RCSB) Protein Databank. Discovery studio 4.5 version (DS, Accelrys Software Inc., USA) was used to perform the process of the removal of water molecules and addition of the missing hydrogen atoms of the AChE enzyme. The highest binding affinity (lowest binding energy) score was selected to explore the binding enzyme–ligand interactions and displayed using the Discovery studio visualization software. Autodock 4.2 was used to dock the AChE enzyme and triterpene compounds (ligands) into the grid box with dimensions of 50 × 50 × 50 Å in the docking option. The best targeted compounds were analyzed according to the binding interactions between ligand and enzyme, such as hydrogen bonding, cation-π, and π-π stacking interactions.

### 3.6. Statistical Analysis

The experiments were expressed as the mean ± standard error of the mean (SEM) in triplicates.

## 4. Conclusions

The present study demonstrated the potential of two extracts of accessions of CA (SECA-K017 and SECA-K018) as potential sources of AChE inhibitors. The results from the in vitro studies indicated the very good interactions of asiatic acid and madecassic acid with the active sites and fulfilled the docking parameters against AChE. Asiatic acid and madecassic acid, which are present in the accessions, showed a favorable AChE inhibitory profile and, therefore, can be used as markers to guide further studies on CA as a potential natural product for the treatment of AD.

## Figures and Tables

**Figure 1 molecules-25-03353-f001:**
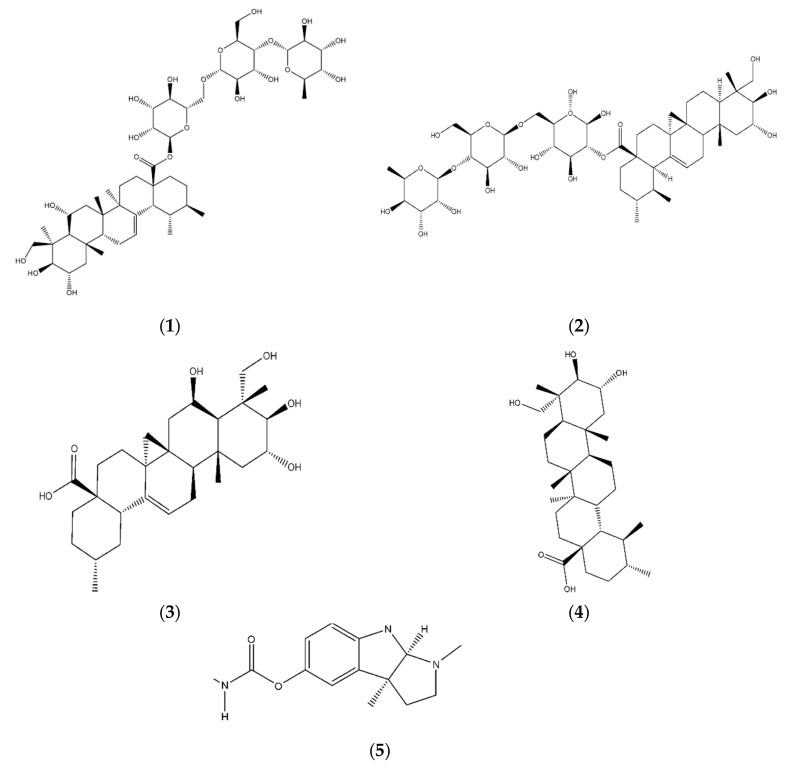
Triterpenes identified in *Centella asiatica* (L.) Urban (CA) (**1**) madecassoside; (**2**) asiaticoside; (**3**) madecassic acid; (**4**) asiatic acid; (**5**) eserine.

**Figure 2 molecules-25-03353-f002:**
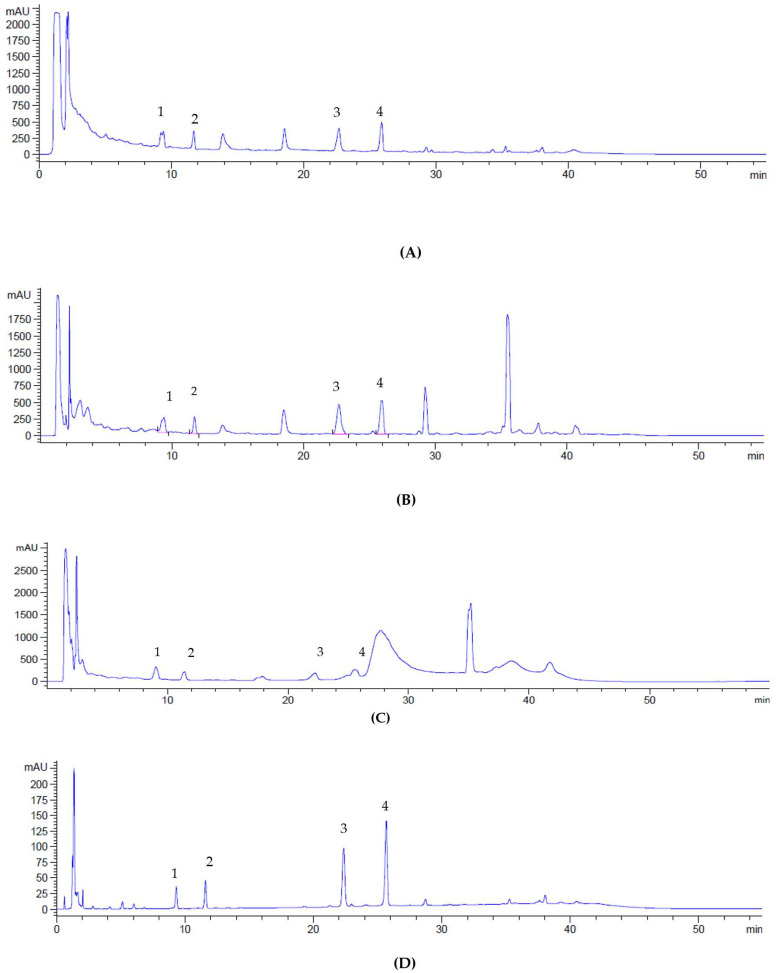
Chromatogram of standardized extracts of CA accessions; SECA-K017 (**A**), SECA-K018 (**B**), SECA-K019 (**C**) and reference standards mixture (**D**). (1) madecassoside; (2) asiaticoside; (3) madecassic acid; (4) asiatic acid.

**Figure 3 molecules-25-03353-f003:**
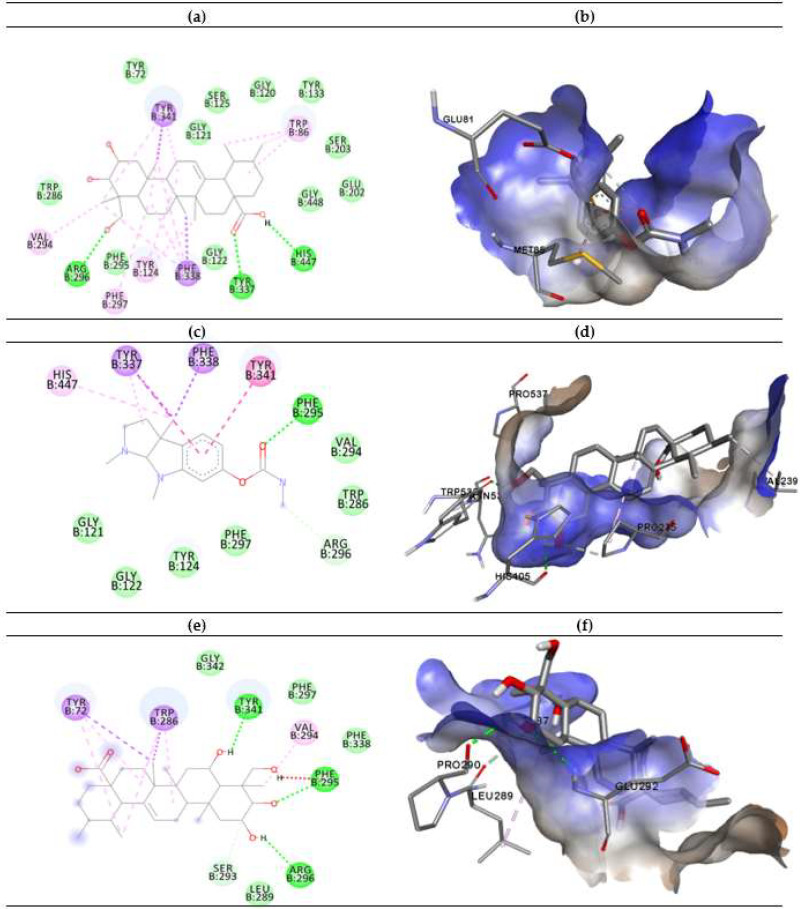
The 2D binding poses representation of compounds (**a**) eserine, (**c**) asiatic acid, and (**e**) madecassic acid with the active sites of AChE. The green dotted lines indicate hydrogen bonds, while π-π stacking and π-sigma interactions are presented in both strong and light magenta between compounds and AChE. The hydrophobic interactions of compounds (**b**) eserine, (**d**) asiatic acid, and (**f**) madecassic acid against AChE.

**Table 1 molecules-25-03353-t001:** Characterization of CA accessions.

Accessions		Characteristics
**CA-K017**	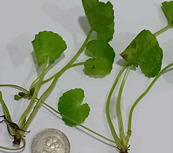	Known as *pegaga kampung* Green colour Glabrous stem Crenate with dentate base Mean leaves diameter: 2.88 ± 0.03 cm
**CA-K018**	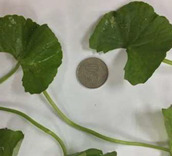	Known as *pegaga daun lebar* Glabrous stem Dark green colour Crenate leaves Mean leaves diameter: 7.57 ± 0.14 cm
**CA-K019**	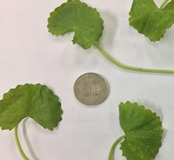	Known as *pegaga nyonya* Glabrous stem Light green colour Crenulate Mean leaves diameter: 4.17 ± 0.05 cm

**Table 2 molecules-25-03353-t002:** Percentage yield of CA accessions extracts.

Accessions	Extracts	Yield of Extraction (%)
CA-K017	SECA-K017	9.80
CA-K018	SECA-K018	56.00
CA-K019	SECA-K019	12.30

**Table 3 molecules-25-03353-t003:** Triterpenes content of standardized extracts of CA accessions.

Triterpenes Compounds	Peak	Retention Time, min	Concentration of Triterpenes in CA Accessions ± SEM (mg/g)		
			SECA-K017	SECA-K018	SECA-K019
Madecassoside	1	9.395	239.23	179.64	252.67
Asiaticoside	2	11.709	190.37	105.71	139.46
Madecassic acid	3	22.679	114.51	112.82	57.88
Asiatic acid	4	25.925	122.1	132.26	103.51

**Table 4 molecules-25-03353-t004:** AChE inhibitory activity of standardize triterpenes and extracts of CA accessions.

Compounds/Extracts	IC_50_ Values (µg/mL)
Madecassoside	37.14 ± 0.04
Asiaticoside	59.13 ± 0.18
Madecassic acid	17.83 ± 0.06
Asiatic acid	15.05 ± 0.05
SECA-K017	481.5 ± 0.13
SECA-K018	763.5 ± 0.16
SECA-K019	>1000
Eserine	0.05 ± 0.12

**Table 5 molecules-25-03353-t005:** Estimated binding energy of triterpenes in the active sites of AChE.

Compounds	Binding Energy (Kcal·mol^−1^)
Madecassoside	81.61
Asiaticoside	41.72
Madecassic acid	−8.7
Asiatic acid	−10.27
Eserine	−9.4

**Table 6 molecules-25-03353-t006:** Gradient condition for HPLC.

Time (min)	Pump A, Water (%)	Pump B, Acetonitrile (%)
0	80	20
15	65	35
30	35	65
35	20	80
40	20	80
45	80	20
55	80	20

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
