# Peer review of "Combining In Silico and In Vitro Studies to Evaluate the Acetylcholinesterase Inhibitory Profile of Different Accessions and the Biomarker Triterpenes of Centella asiatica"

_molecules, 2020, doi:10.3390/molecules25153353_

Round 1

Reviewer 1 Report

The manuscript entitled "Combining in silico and in vitro studies to evaluate the acetylcholinesterase inhibitory profile of different accessions and the biomarker triterpenes of centella asiatica" is describing the inhibitory activity of the triterpenes and centella asiatica (CA) extracts (SECA-K017, SECA-K018, and, SECA-K019) were compared with standard AChE inhibitor eserine. The present study suggested that asiatic acid and madecassic acid in the CA extracts could be responsible for the AChE inhibitory action and can be used as markers to guide further studies on CA as potential natural products for the treatment of Alzheimer’s disease (AD). The findings described here is interesting. It is suggested that the author can be applied to in vitro (cell line) or in vivo (animal model) in further experiments.

Author Response

The manuscript entitled "Combining in silico and in vitro studies to evaluate the acetylcholinesterase inhibitory profile of different accessions and the biomarker triterpenes of centella asiatica" is describing the inhibitory activity of the triterpenes and centella asiatica (CA) extracts (SECA-K017, SECA-K018, and, SECA-K019) were compared with standard AChE inhibitor eserine. The present study suggested that asiatic acid and madecassic acid in the CA extracts could be responsible for the AChE inhibitory action and can be used as markers to guide further studies on CA as potential natural products for the treatment of Alzheimer’s disease (AD). The findings described here is interesting. It is suggested that the author can be applied to in vitro (cell line) or in vivo (animal model) in further experiments.

Answer: 

Noted and thank you

Reviewer 2 Report

This manuscript by Adenan and co-works profiled three accession extracts of Centella asiatica namely SECA-K017, SECA-K018, and SECA-K019 by HPLC, and demonstrated that SECA-K017 and SECA-K018 have moderate acetylcholinesterase inhibitory activity. It provided meaningful results for others in related area. Therefore, I recommend publishing this work after some minor changes.

1) Figure 1, structure (1). Since authors drew other structures with stereo configuration for (2) to (5), structure (1) should also follow this format if there is one.

2) Table 1, CA-K017 has different font style from CA-K018 and CA-K019. Also, table format may be modified for better visualization.

3) Table 3, Madecassoside is bold style, the rest compounds are regular, please make them consistent.

4) Figure2, these 4 HPLC spectra have different retention time frames. These inconsistency can make others hardly compare these 4 triterpenes compounds in different spectra at same time. Moreover, such as spectrum A, all peaks lables (1 to 4) overlayed the peaks. Authors should learn how to insert a text box with "no fill" background in word or other edit software.

5) As a manuscript to discover acetylcholinesterase inhibitors, authors can consider to include more review citations such as Curr Neuropharmacol. 2013 May; 11(3): 315–335.

Author Response

1) Figure 1, structure (1). Since authors drew other structures with stereo configuration for (2) to (5), structure (1) should also follow this format if there is one.

Answer

Structure 1 in Figure 1 (highlighted in yellow) is corrected with stereo configuration as suggested.

2) Table 1, CA-K017 has different font style from CA-K018 and CA-K019. Also, table format may be modified for better visualization

Answer:

The font style of CA-K017 is corrected (unbold). Photo for CA-K017, CA-K018, and CA-K018 are aligned with border

3) Table 3, Madecassoside is bold style, the rest compounds are regular, please make them consistent

Answer:

"Madecassoside" is unbold and consistent with other compounds. 

4) Figure 2, these 4 HPLC spectra have different retention time frames. These inconsistency can make others hardly compare these 4 triterpenes compounds in different spectra at same time. Moreover, such as spectrum A, all peaks lables (1 to 4) overlayed the peaks. Authors should learn how to insert a text box with "no fill" background in word or other edit software.

Answer:

Retention time frame of the 4 HPLC spectra are corrected with similar retention time scale. The peaks labels (1 to 4) were inserted text box with "no fill" background. 

5) As a manuscript to discover acetylcholinesterase inhibitors, authors can consider to include more review citations such as Curr Neuropharmacol. 2013 May; 11(3): 315–335

Answer:

Added information cited from Colovic et al, (2013) (line 56-58, highlighted in yellow) 

Reviewer 3 Report

The article entitled: “Combining In Silico and In vitro Studies to Evaluate the Acetylcholinesterase Inhibitory Profile of Different Accessions and the Biomarker triterpenes of Centella asiatica” is a well explained and easy to follow the paper.

However, I have some minor comments:

  1. Line 112: CA-K019 has a bigger diameter of leaves, so it should be mentioned as the second one, not third
  2. Figure 2: the horizontal scale is not the same size in every plot. This makes it more difficult to compare the position of individual peaks between the plots.
  3. It would be interesting if the Authors would discuss the lack of AchE inhibitory activity of SECA-K019 extract, for example in the light of the HPLC results.
  4. In general SECA AChE inhibitory activity is rather poor. Why Authors did not reconsider plant metabolites extraction for leaves as Aziz 2007 did? Maybe then the concentration of triterpenes would be higher and as a consequence IC50 values of extracts better?
  5. Molecular docking: describing the mechanism of inhibition (including the active site of the enzyme) would make it easier for the reader to interpret the results of the docking (Figure 3).

Lines 215-216: mentioned hydrogen bond between serine and Phe295 is not visible on Fig. 3.

Is this a mistake?

Fig 3 a, c, and e is not so informative, especially for readers more experienced in working with structures. It would be nice if the Authors could add extra figures of structures (even to supplementary data) where the interaction between the most important amino acid residues (represented as sticks) and docked compound is shown. The length of the bond should also be included. Without it, it’s hard to say if the interactions are strong and the Authors suggest that some of them are strong.  The information where the active site of AChE is located in the structures would be welcomed.

  1. 2. How long extraction was carried out?
  2. Line 259 and 265: this is not Table 1,

Author Response

1. Line 112: CA-K019 has a bigger diameter of leaves, so it should be mentioned as the second one, not third

Answer:

As suggested, CA-K019 is mentioned as the second one (line 114, highlighted in yellow. 

2. Figure 2: the horizontal scale is not the same size in every plot. This makes it more difficult to compare the position of individual peaks between the plots.

Answer:

Retention time frame of the 4 HPLC spectra are corrected with similar retention time scale. in addition, the peaks lables (1 to 4) were inserted text box with "no fill" background.

3. It would be interesting if the Authors would discuss the lack of AChE inhibitory activity of SECA-K019 extract, for example in the light of the HPLC results.

Answer: 

Agree with the suggestion of the reviewer. Added information (line 200-201, highlighted in yellow). 

4.In general SECA AChE inhibitory activity is rather poor. Why Authors did not reconsider plant metabolites extraction for leaves as Aziz 2007 did? Maybe then the concentration of triterpenes would be higher and as a consequence IC50 values of extracts better?

Answer: 

Aziz et al, (2007) used methanol as an extraction solvent. We are using ethanol as solvent for future potential pharmaceutical application. We are currently working to improve the extraction procedure to enhance the bioactivity. 

5.Lines 215-216: mentioned hydrogen bond between serine and Phe295 is not visible on Fig. 3.

Is this a mistake?

Answer;

Added new figure S1 (a) to show the interaction between serine and Phe295. 

5. Fig 3 a, c, and e is not so informative, especially for readers more experienced in working with structures. It would be nice if the Authors could add extra figures of structures (even to supplementary data) where the interaction between the most important amino acid residues (represented as sticks) and docked compound is shown. The length of the bond should also be included. Without it, it’s hard to say if the interactions are strong and the Authors suggest that some of them are strong.  The information where the active site of AChE is located in the structures would be welcomed.

Answer:

Extra figures of structures for compounds and amino acid are added (Figure S1, (a), (b), (c)) to indicate interaction between the most important amino acid residues (represented as sticks) and docked compound. The length of the bond is also included in the figure and mentioned in the text (line 226-228, highlighted in yellow). 

 The information on the active site of AChE is located in the structures are added (line 219-222, highlighted in yellow)

6. How long extraction was carried out?

Answer:

72 hours. Added in line 259, highlighted in yellow.

7.Line 259 and 265: this is not Table 1

Answer: 

Table 1 is corrected to Table 6 (line 268 and 274, highlighted in yellow). 
